# Structural insight into the role of novel SARS-CoV-2 E protein: A potential target for vaccine development and other therapeutic strategies

**Manish Sarkar**[1]☯*, **Soham Saha**[2,3]☯*

**1** Department of Biochemistry, Bose Institute, Kolkata, India, **2** Laboratory for Perception and Memory, Institut Pasteur, Paris, France, **3** Centre National de la Recherche Scientifique (CNRS), Unité Mixte de Recherche (UMR-3571), Paris, France

☯ These authors contributed equally to this work.
* sarkarmanish2016@jcbose.ac.in, sarkarmanish2016@gmail.com (MS); soham.saha@pasteur.fr (SS)

**Data Availability Statement:** All relevant data are in the manuscript text in the form of tables, figures and supplementary information. The '.pdb' models of proteins generated are shared in the GitHub

## Abstract

The outbreak of COVID-19 across the world has posed unprecedented and global challenges on multiple fronts. Most of the vaccine and drug development has focused on the spike proteins and viral RNA-polymerases and main protease for viral replication. Using the bioinformatics and structural modelling approach, we modelled the structure of the envelope (E)-protein of novel SARS-CoV-2. The E-protein of this virus shares sequence similarity with that of SARS- CoV-1, and is highly conserved in the N-terminus regions. Incidentally, compared to spike proteins, E proteins demonstrate lower disparity and mutability among the isolated sequences. Using homology modelling, we found that the most favorable structure could function as a gated ion channel conducting $H^+$ ions. Combining pocket estimation and docking with water, we determined that GLU 8 and ASN 15 in the N-terminal region were in close proximity to form H-bonds which was further validated by insertion of the E protein in an ERGIC-mimic membrane. Additionally, two distinct "core" structures were visible, the hydrophobic core and the central core, which may regulate the opening/closing of the channel. We propose this as a mechanism of viral ion channeling activity which plays a critical role in viral infection and pathogenesis. In addition, it provides a structural basis and additional avenues for vaccine development and generating therapeutic interventions against the virus.

## Introduction

The COVID-19 (CoronaVIrus Disease 2019) is a severe acute respiratory syndrome (SARS) caused by a novel coronavirus, SARS-CoV-2, and has taken the form of a worldwide pandemic in the last few months [1]. To date (as in 9 July, 2020), this disease has affected nearly 12 million people, resulting in nearly 550 thousand deaths disrupting social and economic structures of nearly 190 nations across the globe with numbers still on the rise [2, 3].

Coronavirus is a positive-sense single-stranded RNA virus belonging to the class of β-coronaviruses of the family *Coronaviridae*, and affected humans causing different forms of mild

account: https://www.github.com/
SohamSahaNeuroscience/Covid-19_Eprotein_
models.

**Funding:** The author(s) received no specific
funding for this work.

**Competing interests:** The authors have declared
that no competing interests exist.

**Abbreviations:** Å, Angstrom; ALA, Alanine; ARG,
Arginine; ASN, Asparagine; ASP, Aspartic acid;
BFGS, Broyden-Fletcher-Goldfarb-Shanno
algorithm; BLASTp, Basic local alignment search
tool for proteins; CASTp, Computed Atlas of
Surface Topography of proteins; CHARMM,
Chemistry at Harvard Macromolecular Mechanics;
COVID-19, Coronavirus disease 2019; CYS,
Cysteine; E protein, Envelope protein; E-put,
putative E protein; ER, Endoplasmic Reticulum;
ERGIC, ER-Golgi intermediate compartment; GLN,
Glutamine; GLU, Glutamic Acid; GLY, Glycine; H-
bond, Hydrogen bond; hCoV, human coronavirus;
HIS, Histidine; IC, ion channeling; ILE, Isoleucine;
LAV, Live attenuated vaccine; LEU, Leucine; LYS,
Lysine; MERS, Middle-east respiratory syndrome;
MET, Methionine; MMFF, Merck Molecular Force
Field; NMR, Nuclear magnetic resonance; NS1,
Non-structural protein 1; PDB, Protein data bank;
PDZ, Postsynaptic density protein 95 (PSD95)/
Drosophila disc large tumor suppressor (Dlg1)/
Zonula occludens-1 protein (zo-1); PHE,
Phenylalanine; PPI, Protein-protein interactions;
PRO, Proline; RdRp, RNA dependent RNA
polymerase; RMSD, Root Mean Square Deviation;
RNA, Ribonucleic Acid; SARS, Severe Acute
Respiratory Syndrome; SER, Serine; STRUM,
Structure-based stability change prediction upon
single-point mutation; THR, Threonine; TRP,
Tryptophan; TYR, Tyrosine; VAL, Valine.

common cold. Coronaviruses like hCoV-OC43, HKU, and 229E have been responsible for
these diseases [4]. The emergence of human coronaviruses in the 21st century with enhanced
pathogenicity was seen in the case of SARS-CoV-1 mediated infection in 2002, reporting
almost 8000 detected cases worldwide and a mortality rate of 10%. This was followed by
MERS-CoV mediated near-pandemic in 2012 infecting almost 2500 people worldwide with a
mortality rate of 36% [5]. The ongoing novel SARS-CoV-2 pandemic, on the other hand, has a
lower fatality rate in comparison with the previous coronavirus outbreaks, but a higher
human-to-human contagion efficiency than the previous ones [6]. This has led to the spread of
this infection worldwide, across continents, affecting millions.

Coronaviruses have four main structural proteins- (i) Nucleocapsid protein (N), (ii) Spike
protein (S), (iii) Membrane protein (M), and (iv) Envelope protein (E). The E protein is the
smallest of all the structural proteins of 8–12 kDa and is involved in a wide spectrum of func-
tional repertoire [7]. It comprises of three domains: (i) short hydrophilic N-terminus domain
consisting of 7–12 amino acids, (ii) hydrophobic transmembrane domain which is around 25
amino acids long, and (iii) long hydrophilic C terminal region [8–10]. The hydrophobic trans-
membrane domain oligomerizes to form pentameric ion channels having low or no ion selec-
tivity [11–13] and thus might act as a viroporin. Viroporins are small membrane-embedded
proteins present in different pathogenic viruses including SARS-CoV having ion-conducting
properties. They localize on host membrane and help in the viral production and maturation
processes along with its release, which is synergistically regulated by alteration of ion homeo-
stasis of cellular organelles and are ideal therapeutic targets [14]. Elimination of E protein
from mouse hepatitis virus (MHV) and SARS-CoV-1 did not affect viral production signifi-
cantly, but led to a significant reduction in the maturation and release of viral titers [15, 16].
Additionally, deletion of E protein in transmissible gastroenteritis coronavirus (TGEV) and
MERS-CoV led to a replication-competent phenotype which was deficient in propagation [17,
18]. In SARS-CoV-1, E protein deletion (SARS-CoV-1 ΔE) resulted in increased stress and
expression of apoptotic markers in the cells infected compared to wild type [19], pointing
towards decreased infectivity of the virus. NF-κB signaling pathway was implicated in the
decreased inflammation in SARS-CoV-1 ΔE [20]. All these deletions led to viral attenuation in
three animal models, and conferred protection in immunized hamsters and young or old
mice, when infected with the wild type SARS-CoV-1. This points to E protein as a promising
vaccine target against this pathogen [16, 20–22]. The possibility of E protein being a vaccine
candidate has already been explored with the SARS-CoV. A E protein deletion in a SARS-CoV
causing lethal respiratory diseases like SARS or MERS (SARS-CoV-ΔE) was explored as an
attenuated and effective vaccine [21, 23, 24], but it underwent a reversion and became virulent
in cell cultures or *in vivo* [25]. Deletion mutants in the C-terminal regions of the E protein
without disturbing the PDZ binding motif (PDM) led to an attenuated and stable vaccine in
mice [25].

Earlier studies showed that E protein of SARS-CoV-1 forms pentameric structures with ion
conductive pore which might impact host-pathogen interactions [26–29]. Synthetic E protein
showed slight preference in conduction of cations over anions in reconstituted lipid mem-
branes mimicking the ERGIC membrane, and showed no specific selectivity towards cations
[12, 13]. Mutations like N15A and V25F in the protein suppress the ion channeling (IC) activ-
ity [9, 12]. The IC activity is an important determinant of viral pathogenesis. Recombinant
mouse-adapted SARS-CoV-1 viruses with single point mutations at N15A and V25F sup-
pressed its IC activity. However, the virus incorporated compensatory mutations either at the
site of original mutation or near to it upon serial infections, which reverted back their IC activ-
ity. Intriguingly, mice which were infected by either wild type or revertant SARS-CoV-1 rap-
idly lost weight and died. On the contrary, mice which were infected by mutant SARS-CoV-1

with diminished IC activity, recovered from the disease and survived. Moreover, E gene knock-down did not affect viral production, but decreased edema accumulation, the major cause of the fatal acute respiratory distress syndrome (ARDS). Improvement in the integrity of the lungs and proper localization of $Na^+/K^+$ ATPase involved in edema resolution also correlated with the decreased accumulation of edema upon loss of IC activity of E protein. Animals infected by IC activity deficient SARS-CoV-1, showed reduced inflammasome activated IL-1β levels in the lung airways. The reduced IL-1β levels corroborated with the decrease in the levels of TNF and IL-6. This shows that E protein IC activity mediates the cytokine storms which leads to progressive lung damage and finally ARDS. Thus, the IC activity of E protein is an important and enigmatic determinant of SARS-CoV-1 pathogenesis [11]. E protein is highly conserved across the phylogeny including in SARS-CoV-1 and the new deadly pathogen SARS-CoV-2. So, SARS-CoV-2 pathogenesis can be correlated with the IC activity of its E protein counterpart from SARS-CoV-1.

A β-coil-β motif present in the C terminal region of E protein contains a conserved proline residue [30] which is indispensable for its localization in the ER-Golgi complex [31]. During the viral replication cycle, the virus incorporates a small proportion of the viral E protein expressed within the infected host, into the virion particles [32]. The larger proportion of E protein is localized at the sites of mammalian intracellular trafficking, the ER-Golgi network and the ERGIC [33]. This localization in the intracellular trafficking components of the host cell helps in the assembly of viral structure and budding [33].

Since CoV E-protein forms pentameric non-selective ion channels in the membrane [11–13], we modeled the SARS-CoV-2 E protein using the SARS-CoV-1 NMR structure [10] as the template. M2 viroporin of Influenza A virus has been reported to function as a proton channel with a water-hopping mechanism in a conformational dependent manner [34]. Similarly, we identified key amino acid residues lining the inner luminal side of the SARS-CoV-2 E protein, and subsequent water docking and membrane insertion followed by morphing of the E protein models suggest that the pentameric structure is able to form dynamic closed and open states. While it calls for detailed studies, mutations of the E-protein in the transmembrane region keeping the PBM intact, presents a potential target for the development of vaccines (LAVs or inactivated) and other therapeutic strategies to curb the COVID-19 pandemic.

## Materials and methods

### Multiple sequence alignment

In order to assess the conservation among candidate coronavirus E-proteins, we performed a protein-protein BLAST (BLASTp: Basic Local Alignment Search Tool) with the E protein of SARS-CoV-2 isolated from Shanghai in April (accession number in QII57162.1). Using a 90–100% sequence identity as filters, we aligned the resulting sequences using the online alignment tool Clustal Omega [35] and MegaX [36]. Both the software enables high fidelity multiple protein sequence alignment. Jalview (www.jalview.org) was used for visualizing the sequence alignment, conservation score, quality of alignment and consensus residue for each position. The same methods were undertaken for spike proteins (100 sequences), M proteins (54 sequences) and protein 7a (30 sequences).

### Disparity score and mutability

Estimates of evolutionary divergence are shown for the different E-proteins isolated from different sources (bat (HKU3-7), human SARS 2018, bat SARS_RsSHC014, BtRI-BetaCoV, SARS-CoV-1 and SARS-CoV-2 (accession number: QII57162.1)). Analyses were conducted using the Poisson correction model. All ambiguous positions were removed for each sequence

pair (pairwise deletion option). The disparity index is a measure of heterogeneity among sites [37] and was calculated from MegaX [36]. Briefly, the disparity index uses a Monte-Carlo procedure to test homogeneity among the input sequences. Values greater than 0 indicate the larger differences in amino acid composition biases than expected, based on the evolutionary divergence between sequences and by chance alone.

Frequencies of different amino acids were calculated across the aligned sequences. Mutability was defined as the probability of an amino acid to be non-consistent across the sequence and was calculated as follows:

$$p(M) = 1 - AA(\%)/21$$

where AA: amino acid composition in percentage and 21 indicates the total amino acid residue space. The same analytical methods for spike proteins were performed. All data is represented as mean ± SD.

## Homology modeling of the E-protein

The structure of the envelope (E) protein of SARS-CoV-2 was modeled with a template-based homology modeling approach using the NMR structure of E protein from SARS-CoV-1 (PDB id: 5X29; [10]). The full-length pentameric model of the E protein was generated using the following steps:

1. Backbone and side-chain modeling of the monomeric unit.

2. Loop refining of the monomeric unit.

3. Pentamer modeling from the monomer unit.

4. Model refinement and further structural fine-tuning of unreliable structural regions, using the GalaxyWEB server [38].

Another pentameric model of the E protein was generated using the same template (PDB id: 5X29) using the SWISS-MODEL [39, 40]. The structures obtained from these modeling servers were validated by scores obtained from the MolProbity [41, 42]. The structures were chosen by comparing predominantly the different parameters like percentage of Ramachandran favored and unfavoured residues, percentage of favored and unfavored rotamers, MolProbity score, and Clash Score, validating the quality of the modeled protein.

## Structure visualizations

The structural visualizations of the different models were performed using PYMOL. In addition, we used UCSF-CHIMERA [43] for detailed visualization of residues in the structure and for representation throughout the paper. UCSF-CHIMERA has been referred to as CHIMERA throughout the article. Important amino acid residues lining the inner lumen of the pentameric *E-put* protein were identified. Of particular interest was the PHE 26 where we defined the bottleneck region and this was used to calculate the pore radius for different conformations. The radius of the bottleneck region of the *E-put* protein was calculated with the centroid of the triangle formed by the different orientations of PHE 26 in closed or open conformation. The distance from the vertices of the triangle thus formed from the centroid was then used to calculate the effective radius.

For a scalene triangle having side lengths of a,b and c units, the length of the medians are defined as $m_a$, $m_b$ and $m_c$ units on the respective sides as mentioned in the subscript. The

length of the medians are:

$$m_a = \frac{\sqrt{2b^2 + 2c^2 - a^2}}{2}$$

$$m_b = \frac{\sqrt{2a^2 + 2c^2 - b^2}}{2}$$

$$m_c = \frac{\sqrt{2a^2 + 2b^2 - c^2}}{2}$$

All the medians meet at a point inside the triangle, called the centroid. The distances of the three vertices from the centroid are equal to two-thirds of the length of the median passing through the respective vertices. The distances of each of the vertices from the centroid are considered as the pore radii of the protein in three different directions. The average value of all these radii is calculated as the effective pore radius of the protein.

### Pore volume and topology estimation

The pore volume of the protein in its different conformational states is determined using CASTp (Computed Atlas of Surface Topography of proteins; [44]). In order to account for the presence and impact of the modeled pentameric *E-put* protein, we estimated the presence of surface pockets, internal cavities, and cross channels using the CASTp server. Briefly, CASTp uses a computational geometry approach to measure area and volume and to identify topological imprints on the protein. The lining residues are annotated from the UniProt database and mapped along with the cavity. The cavities with the two maximum molecular surface area and volume were selected. The molecular surface is colored by its hydrophobicity index. These values are determined by the Kyte-Doolittle scale of hydrophobicity with appropriate colors ranging from blue for the most hydrophilic to orange-red for the most hydrophobic pockets. The Ramachandran plots of the protein are plotted using Zeus software (http://www.al-nasir.com/portfolio/; Source: https://www.rcsb.org/pages/thirdparty/molecular_graphics).

### Water molecule interactions in different conformations

The docking of the water molecules to the *E-put* structure was performed in SWISS-DOCK [45].

**Algorithm.** EADock DSS [46] blind docking algorithm is used by SWISS-DOCK and CHARMM [47] force fields are used to create a rank of favorable energies. The most favorable clusters are then used to generate the solvation model.

**Input files.** The modeled protein structure (*E-put*) and ligand (water) were uploaded in the PDB and Mol2 formats respectively. The water molecule (Pubchem ID: CID 962) was imported in CHIMERA in .sdf format and converted to .mol2 format. The SWISS-DOCK server uses CHARMM topology, parameters, and coordinates for determining Merck Molecular Force Field (MMFF) and Van der Waals interactions from the ligand to predict the docking sites on the protein structure.

### Lipid interaction using Autodock Vina

Phosphatidylcholine (Pubchem ID: CID 6441487) and ceramide (Pubchem ID: CID 5702612) are the main lipid components of the ER-Golgi network and ERGIC of mammals including humans. *E-put* has been docked to both the lipid molecules by the following steps:

1. Preparation of lipid molecules in OpenBabel which is an online interface for the preparation of chemical molecules [48].

2. Energy minimization of the modeled structure in CHIMERA using the following steps:

    a. *Fast process*: Steepest descent minimization to remove the critical steric clashes which are highly unfavorable to the protein structure globally.

    b. *Slow process*: Conjugate gradient minimization to remove remaining clashes and other steric discrepancies in the protein structure to reach an energy minimum for that structure.

    c. The energies are reported in kJ/mol in the Reply Log [43].

3. The minimized structure is visualized in CHIMERA and docked to the two lipid molecules using the following steps:

    a. The search space for the docking algorithm to run is defined by a three-dimensional box with appropriate axes' lengths which encompasses our region of interest in the protein structure.

    b. Docking of the lipid molecules to the E protein in the search space described previously is carried out using Autodock Vina which uses the Broyden-Fletcher-Goldfarb-Shanno (BFGS) algorithm as an Iterated Local search global optimizer method, to speed up its optimization procedures [49].

The docked conformations of the protein with the lipid molecules were visualized using CHIMERA and the RMSD values of the docking events analyzed by Vina, are represented by CHIMERA in a Reply Log of Autodock Vina [43, 49].

## Membrane insertion using CHARMM GUI

CHARMM GUI is an interactive platform to generate membrane protein complexes by providing user-defined parameters in each and every step of the entire process [50, 51].

The process consisted of the following steps:

Step 1: The PDB file of the protein was loaded.

Step 2: The protein was oriented along the Z-axis, the principal axis in the case of homo-pentamers. Then the pore water was generated and the cross-sectional area across the entire pore lumen calculated.

Step 3: The system size was determined by using the cross-sectional areas of the protein determined in step 2 and from a library of experimentally derived cross-sectional areas of various lipid molecules. This helped to build a system, using a specific number of lipid-like pseudo atoms which mimics the lipid headgroups, with user-defined dimensions and shapes in the XY plane.

Step 4: The system size information from step 3 was used to build the system components. The pseudo atoms were replaced by lipid molecules as per user-defined parameters such as the number of each lipid component and optimizing them as in Table 1. The pore water generated in step 2 was refined until no more water molecules are present in the membrane occupied hydrophobic regions. Ions like K+ and Cl- are added to the system using a distance-based ion replacement algorithm.

**Table 1. Table showing composition and optimization of the generated membrane.**

|  | ER% | Golgi % | (ER+Golgi) | % in ERGIC complex | N = 500 | N = 450 | N = 400 | N = 350 | N = 300 | N = 250 | N = 200 |
|---|---|---|---|---|---|---|---|---|---|---|---|
| POPC | 54 | 36 | 90 | 47 | 235 | 212 | 188 | 164 | 141 | 118 | 94 |
| POPE | 20 | 21 | 41 | 21 | 105 | 118 | 84 | 74 | 63 | 52 | 42 |
| POPI | 11 | 12 | 23 | 12 | 60 | 54 | 48 | 42 | 36 | 30 | 24 |
| POPS | 6 | Not present | 6 | 3 | 15 | 14 | 12 | 11 | 9 | 7 | 6 |
| Cholesterol | 8 | 18 | 26 | 14 | 70 | 63 | 54 | 49 | 42 | 35 | 28 |
| SM | 6 | Not present | 6 | 3 | 15 | 13 | 12 | 11 | 9 | 7 | 6 |

Step 5,6: The entire system that was assembled, is equilibrated gradually by applying different restraints like (i) planar restraints to position the lipid head groups parallel to the Z-axis (ii) harmonic restraints to the ions and any heavy atoms present in the protein and (iii) repulsive planar restraints that prevent water molecules to enter into the membrane occupied hydrophobic region of the system. Due to the lack of computational power, the CHARMM GUI provided six different steps of equilibration at all the individual steps. The steps 1 and 2 were performed by NVT (constant volume and temperature) dynamics and the rest of the four steps are by NPAT (constant pressure, area, and temperature) dynamics at 303.15 K. This particular difference was applied to stabilize the dynamic integrations in between the components of the system during the equilibration process.

## Morphing

In order to gain an insight into the structural intermediates the *E-put* protein might be undertaking, we resorted to generating the trajectories of change given the two conformations of the protein: open and closed. We employed the "Morphing" functionality in UCSF-CHIMERA. Following pre-alignment of the two conformations using the sequence alignment tool from residue 21–41, the settings used for morphing the structures were- Interpolation rate: linear; Interpolation type: cockscrew; Interpolation steps: 60; Core fraction 0.5. Morphing generates a directed trajectory for two intermediates, thereby providing snapshots of possible changes. Morphing avoids chemical distortions during theoretical chemical transitions like steric hindrances and improbable geometry of covalent bonds [52], thus providing an energy-dependent intermediate model. A physically reasonable morph trajectory considers the high-dimensional nature of macromolecules and thermal motions of the structural ensemble (like the *E-put*) and connects such states. Morphing generates a clash-free, physically rational, trajectory connecting an initial (e.g. open) to a final (e.g. closed) conformation [52].

## Effect of point mutations using ProtParam and STRUM

The physicochemical properties of the protein like theoretical pI, aliphatic index and overall average of hydrophobicity, were found using the ProtParam tool of the ExPasy server (https://web.expasy.org/docs/expasy_tools05.pdf) as in Table 3 using the protein sequence as the input. Point mutations were introduced in the PHE 4, GLU 8, ASN 15 and PHE 26 residues, which are putatively important for the functionality of the protein. We used the STRUM online mutation stability change predictor to assess the effects of the point mutations on the stability of the protein [53]. In brief, STRUM is a machine learning-based mutation stability change predictor that can qualitatively predict the stability of proteins. It is done by a simple two-state model (folded and unfolded), where the stability is defined by the difference in

Gibbs free energy between the unfolded and the folded states. The higher the difference in the free energy between the two states (ddG), the protein could be predicted to be stable against denaturation. With the introduction of a mutation, the free energy landscape and the stability can change, where the difference in the free energy between wild type and the mutant is a measure of how mutation affects the stability. A negative ddG indicates that the mutation is destabilizing for the protein structure [53] and thus expected to be functionally compromised.

## Results

### Coronavirus E-proteins are conserved and have lower mutability compared to spike proteins

A BLASTp search run resulted in 54 sequences of E-proteins from different origins (Fig 1A). E protein of SARS-CoV-2 isolated in April (Accession number: QII57162.1) was used as the search reference. Some E-proteins specific to bat (HKU3-7), human SARS 2018, bat SARS_RsSHC014, bat BtRI-BetaCoV, human SARS-CoV-1 and human SARS-CoV-2 (Accession number: QII57162.1) show a high degree of conservation in their amino acid composition (Fig 1B). In comparison, we also looked at the conservation of amino acids for the spike proteins (S1A Fig), M-proteins (S1B Fig) and protein 7a (S1C Fig). The candidate E-proteins from different origins were closely associated when we plotted the evolutionary distance (S1D Fig). All the selected sequences of Spike proteins, M proteins, E-proteins, and protein 7a were selected based on a 90–100% sequence identity. The amino acid composition indicated that the E-proteins have a larger proportion of Leucine (LEU) and Valine (VAL) compared to either the Spike or the M-proteins (Fig 1C-i-iii). Of particular interest was the increased disparity index among spike proteins compared to the E-proteins (Fig 1D-i-ii). The value of the disparity index close to 1 (red) indicated larger differences in amino acid composition bias than expected only out of chance probability and evolutionary divergence [26]. Using the amino acid composition, we calculated the probability of accumulating mutation for a particular residue as a proxy of mutability. We found that leucine (LEU, $0.189 \pm 0.132$), valine (VAL, $0.25 \pm 0.13$) and serine (SER, $0.461 \pm 0.091$) (Fig 1E; indicated by $$) of the E proteins have lower than 50% chance of mutability compared to spike proteins (LEU ($0.626 \pm 0.004$), VAL ($0.652 \pm 0.004$) and SER ($0.632 \pm 0.006$)) (Fig 1E). Given the role of E proteins in viral infection [19], we think that E-proteins present to us a relatively stable viral protein that demands greater investigation to elucidate its mechanistic insights and features.

### Homology modeling of SARS-CoV-2 E protein, validation and bottleneck measurements

We designed a template-based pentameric structural model of the E protein of SARS-CoV-2 with the NMR structure of E protein from SARS-CoV-1 (PDB ID: 5X29; (10)) as the template (Fig 2A, side view; Fig 2C, top view). The putative pentameric models of the E-protein have been obtained from GalaxyWEB server (Fig 2B-i, side view, Fig 2D-i, top view) and SWISS-MODEL (Fig 2B-ii-side view, Fig 2D-ii-top view). Based on homology modeling, we obtained 16 different structural models from both the GalaxyWEB (S2A–S2P Fig) and SWISS-MODEL (S3A–S3P Fig). The model obtained from GalaxyWEB spanned the entire protein (1–75) but the five residues (PDLLV) in the C-terminal end of the protein were removed which reduces the structural flexibility of that region improving the MolProbity parameters. The model obtained from SWISS-MODEL spanned from 8–65 residues similar to the NMR structure. All the models from both these protocols and the template structures were validated using Mol-Probity. The GalaxyWEB models were chosen based on the following observations: (i) they

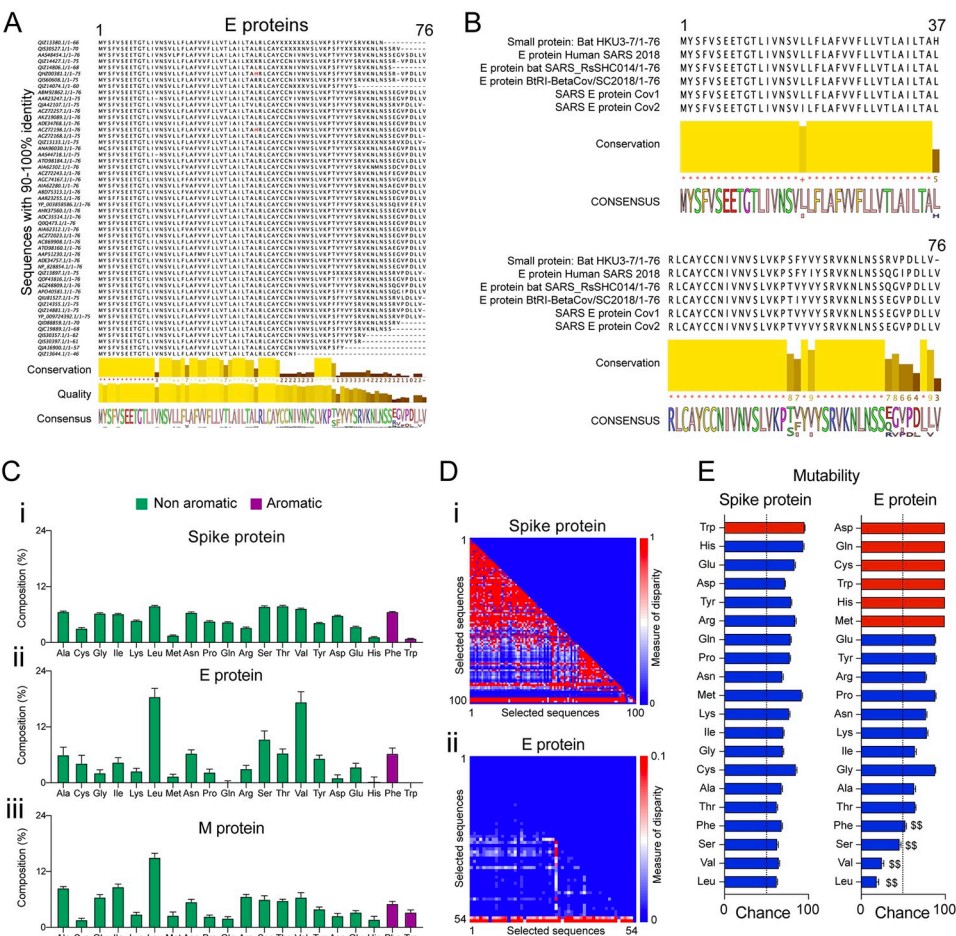

**Fig 1. Sequence alignment of SARS CoV E proteins, disparity index and mutability.** (A) Sequence alignment of CoV E-proteins performed by Clustal Omega after a BLASTp search against human SARS-CoV-2 E protein (Accession number: QII57162.1). The selected aligned sequences were based on the criteria of 90–100% similarity. The conserved sequences, quality of sequence alignment, and consensus sequence are shown below. (B) Sequence alignment of CoV E-proteins from the bat (HKU3-7), human SARS 2018, bat SARS_RsSHC014, BtRI-BetaCoV, SARS-CoV-1 and SARS-CoV-2 showing highly conserved regions in the N-terminal region. The consensus sequence is shown below. (C) Percentage composition of each amino acid computed from the aligned selected sequences for (i) Spike protein, (ii) E-protein, and (iii) M-protein. (D) Lower triangular heatmap representation of the disparity index computed from the sequence alignment for (i) spike protein, and (ii) E-protein. Scale: 0–1 (for i), and 0–0.1 (for ii). (E) Mutability (probability of amino acid change) calculated for spike protein and E-protein. The dotted line indicates a 50% probability chance, while red bars indicate amino acids absent/single-site presence on the sequences analyzed. $$ indicates residues with lower mutability in E-protein compared to spike protein.

spanned a longer region of the protein (Fig 2G-i and 2Gii) and (ii) they had better validation scores (Clash score and MolProbity score). The final structural model (Fig 2B-i) was chosen depending on the structural validation parameters obtained from MolProbity (Fig 2H-green). We refer to the structural model used here for further characterization as "*E-put*". The bottle-neck radius of the template structure (4.2Å; Fig 2E; Table 2) and the models obtained from GalaxyWEB (3.4Å; Fig 2F-i; Table 2) and SWISS-MODEL (3.4Å; Fig 2F-ii; Table 2) were cal-culated using the centroid formed by the triangular orientation of the PHE 26 residues (Fig 2H; refer to *Methods*).

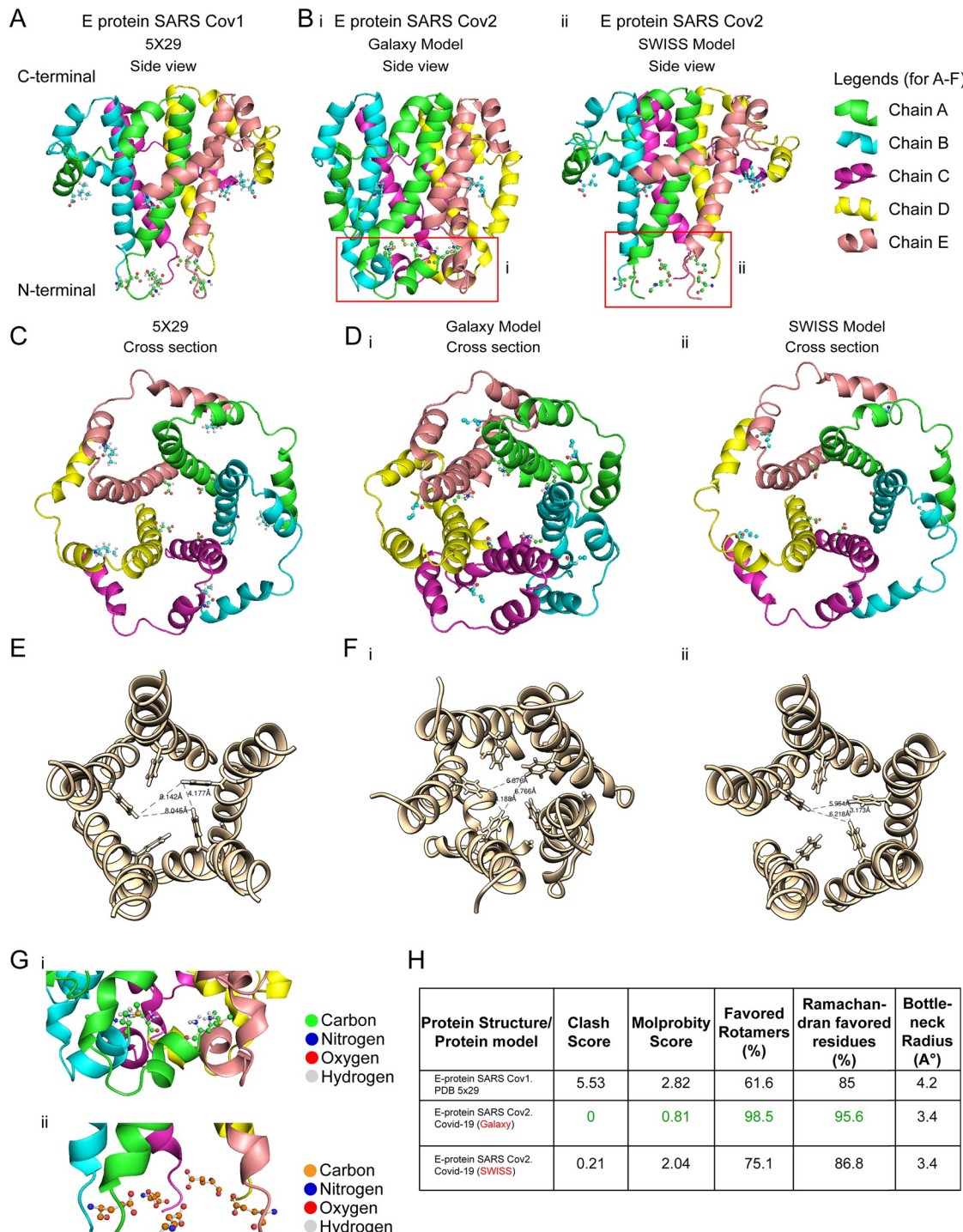

**Fig 2. Pentameric homology model of the E protein of SARS-CoV-2.** (A) NMR structure of pentameric E protein (PDB id: 5X29) of SARS-CoV-1. (B) Pentameric model of E protein of SARS-CoV-2 (COVID-19) (i) generated in GALAXY (*E-put*), and (ii) generated in SWISS-MODEL. A magnified figure for each is provided in (red box). (C) Top view of pentameric E protein SARS-CoV-1 (5X29). (D) Top view of pentameric E protein SARS-CoV-2 (i) generated in GALAXY (*E-put*), and (ii) generated in SWISS-MODEL. (E) Bottleneck radius measurement of C (5X29). (F) Bottleneck radii measurement of D (i) GALAXY (*E-put)* and (ii) SWISS-MODEL. (G) N-terminal residues of (i) GALAXY model (*E-put*) (as in B-i) and (ii) SWISS-MODEL model (as in B-ii). (H) Structure validation parameters and pore radii of 5X29, GALAXY model (*E-put*), and SWISS-MODEL model.

**Table 2. Table showing MolProbity score, clash score, and pore radius for three possible conformations of the *E-put* protein.**

| Models | MolProbity Score | Clash Score | Pore radius (angstrom) |
|---|---|---|---|
| E-protein SARS-Cov 2 (Galaxy model) | 0.81 | 0 | 3.4 |
| E-protein SARS-Cov 2 (Galaxy model: open; S3C Fig) | 2.79 | 15.66 | 5.6 |
| E-protein SARS-Cov 2 (Galaxy model: closed bottleneck; S3H Fig) | 2.02 | 12.86 | 2.3 |

## *In silico* structural analysis of *E-put* showing water docking and putative lining residues important for its functionality as an ion channel conducting H⁺ forming H-bonds with water

The pore volume of the *E-put* protein was calculated using the CASTp server clearly showing a discontinuous channel in the luminal surface of the pore consisting of Pocket 1 and Pocket 2 (Fig 3A). The lining residues were LEU 37, ILE 33, VAL 29, PHE 26, ALA 22, LEU 19, LEU 18, ASN 15, GLU 8, VAL 5 and PHE 4 (Fig 3B-i). The luminal surface of the pore is shown according to its hydrophobicity content (Fig 3B-ii). The presence of pockets lined by charged, aromatic and hydrophobic residues suggested that the pentamer could possibly form continuous water chains for H⁺ ion channeling [34]. To gain an insight into this hypothesis, we docked *in silico* water molecules with the *E-put* protein using SWISS-DOCK. The presence of water molecules docked in the N-terminal end was observed (Fig 3C). The distances of the marked residues (LEU 37, ALA 36, ILE 33, ASN 15 and GLU 8) from their nearest water molecules showed that ASN 15 and GLU 8 lie in the range of hydrogen bonding limit (<2.5 Å; Fig 3D). As expected, the distance of LEU 37, ALA 36, ILE 33 from the nearest water molecules lie beyond the hydrogen bond limit and also short-range interactions limit (>3.8 Å; Fig 3D).

The structural orientation of the ASN 15 and GLU 8 of the *E-put* protein made these residues ideal candidates for water docking (Fig 3E-i-ii, left, side view, S4A-i-ii Fig, top view). The distance from a dimer water molecule docked in the luminal side of the pore to the carbonyl group of the amide side chain of ASN 15 and the charged carboxyl group of the side chain of GLU 8 was 2.03Å and 1.94Å respectively (Fig 3E-i-ii, right). Both were seen to be perfectly within conventional H-bonding limits.

The structural orientation of LEU 18 and ALA 22 along with their side chains formed the 'central core' (Pocket 1) of the *E-put* protein (Fig 3F, left, side view, Fig 3F, right, top view, S4B Fig, top view). The LEU 37, ALA 36 and ILE 33 residues along with their side chains formed the 'hydrophobic funnel' (Pocket 2) of the *E-put* protein (Fig 3G, left, side view, S4C Fig, top view). The distance from a trimer of water molecules docked in the luminal side of the pore to the hydrophobic side chains of the comprising residues was much higher than H-bonding limit (2.5 Å) and short-range interactions limit (3.8 Å) (Fig 3G, right; LEU 37–5.39 Å, ALA 36–5.33 Å, ILE 33–5.91 Å). This shows that water molecules can access the interior of the hydrophobic funnel without any unfavorable interactions or steric clashes.

The structural orientation of PHE 26 and their side chains formed the bottleneck of the *E-put* protein, which can potentially limit the conduction of water or ions through the channel (Fig 3H, left, side view, Fig 3D, right, top view). The structural orientation of PHE 4 forming the gate of the *E-put* protein, the conformation of which could determine the putative closed (Fig 3I, left, side view, Fig 3I, right, top view) and open states (S4D Fig, left, side view, S4D Fig, right, top view).

Simultaneously, we analyzed the model obtained from SWISS-MODEL (Fig 2B-ii and 2D-ii). CASTp pocket analysis clearly showed the discontinuous channel in the luminal surface of the pore consisting of Pocket 1 and Pocket 2 (Fig 4A), and the lining residues (Fig 4B-i). The hydrophobicity content is similarly visible on the luminal surface (Fig 4B-ii). We docked

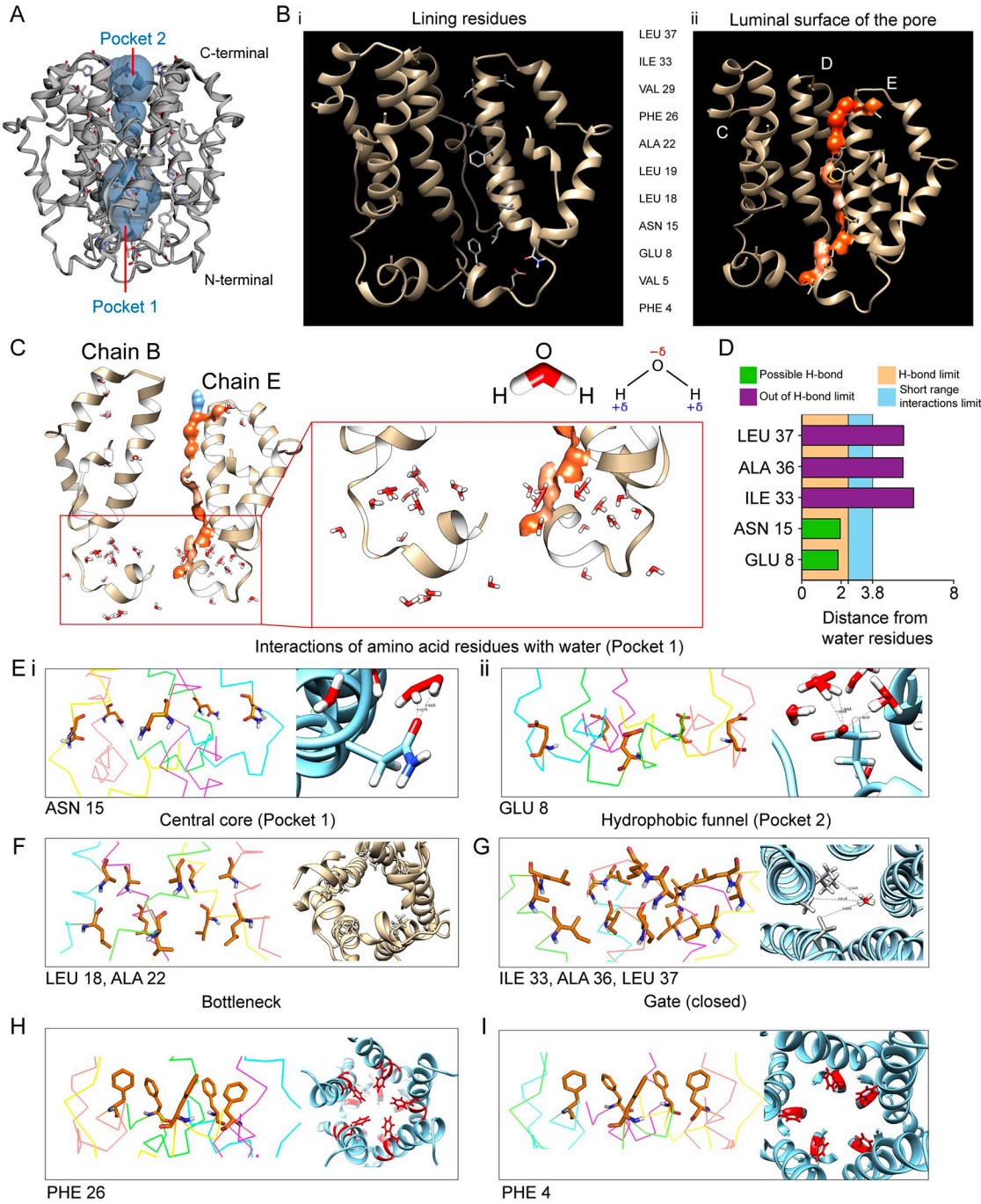

**Fig 3. Pore volume estimation of *E put* protein by GALAXY-WEB and docking with water molecules.** (A) Estimation of the pore volume of the GALAXY-WEB *E-put* protein using the CASTp 3.0 server. (B) (i) Residues lining the luminal side of the *E-put* protein core and (ii) the luminal surface of the pore determined by the CASTp 3.0 server. (C) Docking of water to the *E-put* protein by SWISS-DOCK showing the chains B and E of the pentamer and magnification of the region of interaction. (D) Distances of water molecules from different lining residues and the limit of H-bond. (E) Residues' orientation of the *E-put* protein generated from PYMOL and interaction of water with residues showing the distance between water and (i) ASN 15 and (ii) GLU 8 by CHIMERA. (F) Residues' orientation generated from PYMOL (side view) and CHIMERA (top view) lining the Central core of the *E-put* protein. (G) Residues' orientation generated from PYMOL and interaction of water with the lining residues of the Hydrophobic funnel of the *E-put* protein by CHIMERA. (H) Residues' orientation generated from PYMOL (side view) and CHIMERA (top view) showing the Gate of the *E-put* protein in the closed conformation. (I) Residues' orientation generated from PYMOL (side view) and CHIMERA (top view) forming the bottleneck of the *E-put* protein.

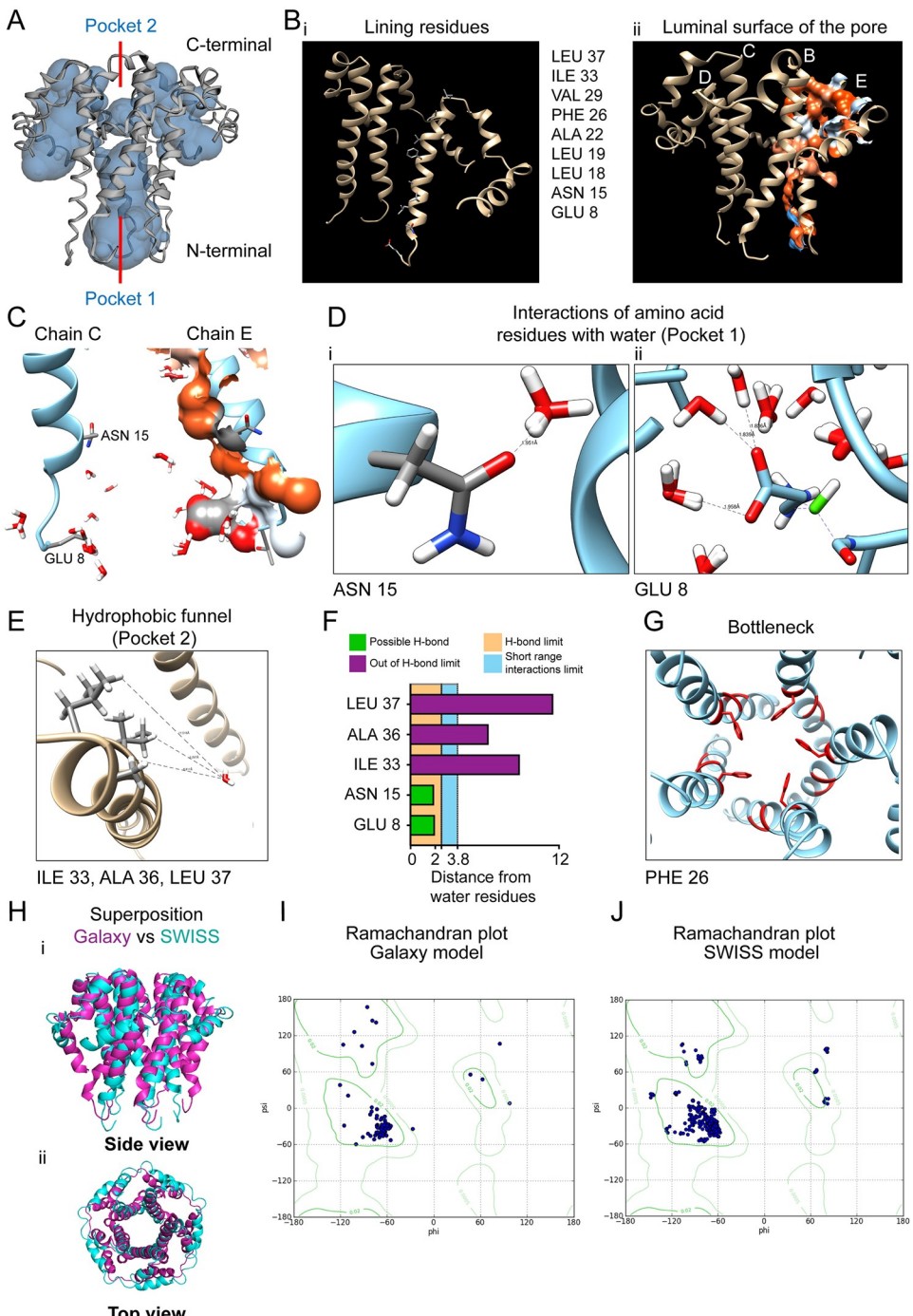

**Fig 4. Pore volume estimation of *E put* protein by SWISS MODEL and docking with water molecules.** (A) Estimation of the pore volume of the SWISS-MODEL *E-put* protein using the CASTp 3.0. (B) (i) Residues lining the luminal side of the SWISS-MODEL protein core and (ii) the luminal surface of the pore determined by the CASTp 3.0 server. (C) Docking of water to the SWISS-MODEL protein by SWISS-DOCK showing the chains C and E of the pentamer. (D) Residues' interaction of water with residues showing the distance between water and (i) ASN 15 and (ii) GLU 8 by CHIMERA. (E) Residues' interaction of water with the lining residues of the hydrophobic funnel of the SWISS-MODEL protein by CHIMERA. (F) Distances of water molecules from different lining residues and the limit of H-bond. (G) Residues' orientation in CHIMERA forming the bottleneck of the SWISS-MODEL protein. (H) Superposition of the *E-put* and the SWISS-MODEL protein (purple: *E-put*; cyan: SWISS-MODEL) as (i) side view and (ii) top view. (I) Ramachandran plot of the *E-put* protein (Galaxy model). (J) Ramachandran plot of the SWISS-MODEL protein.

*in silico* water molecules with the SWISS-MODEL protein using SWISS-DOCK (Fig 4C). Similar to E-put shown above, the distances of ASN 15, and GLU 8 from their nearest water molecules showed that they lie in the range of hydrogen bonding limit (<2.5 Å; Fig 4D-i and 4Dii), while LEU 37, ALA 36, ILE 33 lie beyond the hydrogen bond limit and also short-range interactions limit (>3.8 Å; Fig 4E). The distance from a dimer water molecule to the carbonyl group of ASN 15 and the charged carboxyl group of GLU 8 was 1.95 Å and 1.88 Å respectively (Fig 4F). The PHE 26 and their side chains formed the bottleneck of the SWISS-MODEL protein (Fig 4G).

The modeled *E-put* protein from GalaxyWeb (Fig 2-b-i) and SWISS-MODEL (2-b-ii) was superimposed using Pymol which yielded an RMS (RMSF) of 3.865 (Fig 4H-i, side view, Fig 4H-ii, top view). In addition, we superimposed the *E-put* obtained from the GalaxyWeb and SWISS-MODEL with the reported NMR CoV-1 E protein structure (S5A and S5B Fig: i (top view), ii (side view). Ramachandran plot of the *E-put* obtained from the GalaxyWeb and SWISS-MODEL showed that most of the favored residues of both the models fall in the alpha-helical region (Fig 4I and 4J).

## Docking of *E-put* with lipid molecules: Ceramide and phosphatidylcholine

Ceramide and phosphatidylcholine contribute to the maximum percentage of lipid composition in the ER-Golgi network and ERGIC complex of the mammalian cells including humans. Ceramide (Fig 5A) and phosphatidylcholine (Fig 5D) were docked to *E-put* protein using CHIMERA and Autodock Vina (Fig 5B and 5E respectively; refer to *Methods*). The RMSD values for both the cases were zero, validating the quality check of the lipid-bound structure of *E-put*. The distances of the terminal guanidino group of the side chain of ARG 38 with the phosphoryl group of the charged headgroup of ceramide (2.32 Å; Fig 5C) and the carbonyl group of the charged headgroup of phosphatidylcholine (2.29 Å, Fig 5F) were calculated. The distances fall within the covalent or strong hydrogen bond limit (Fig 5G). This concludes that the *E-put* protein docks to the lipid components of the ER-Golgi network and ERGIC complex effectively either by covalent or strong hydrogen bond formation.

## *In silico* structural analysis of *E-put* shows a putative conformation-dependent proton channeling mechanism

The mechanism of proton transport through the M2 channel of influenza A virus *via* the formation of well-ordered clusters of water molecules has already been reported [34]. This protein undergoes a pH-dependent conformational change between its closed and open state ([34]; Fig 6A). Based on the M2 channel mechanism and our findings, we propose that *E-put* protein undergoes conformational changes: (1) Closed discontinuous state (Figs 2B-i and 6B-i, top- lateral view, bottom- cross-section) to (2) Continuous channel state 1 (Fig 6B-ii top- lateral view, bottom- cross-section; S2H Fig) to (3) Continuous channel state 2 (Fig 6B-iii, top-lateral view, bottom- cross-section; S2J Fig) and back to Closed discontinuous state (Figs 2B-i and 6B-iv, top- lateral view, bottom- cross-section). The pore volumes of all the putative states of *E-put* respectively (Fig 6C) showed an increase in pore volume during the transition from the closed discontinuous state (Fig 6C-i) to continuous channel states 1 and 2 (Fig 6C-ii-iii). The volume finally decreases when the conformation returns back to the closed discontinuous state (Fig 6C-iv). The change of pore volume in these putative intermediates also corroborates with the opening and closing mechanism of *E put* (Fig 6D). The water docking phenomena performed in SWISS-DOCK with all the three putative conformational states further validates this hypothesis (S7A-i-iii Fig). As a molecular mechanism of how these physical states might be achieved, the orientations of the bottleneck residue PHE 26 in all the three states were

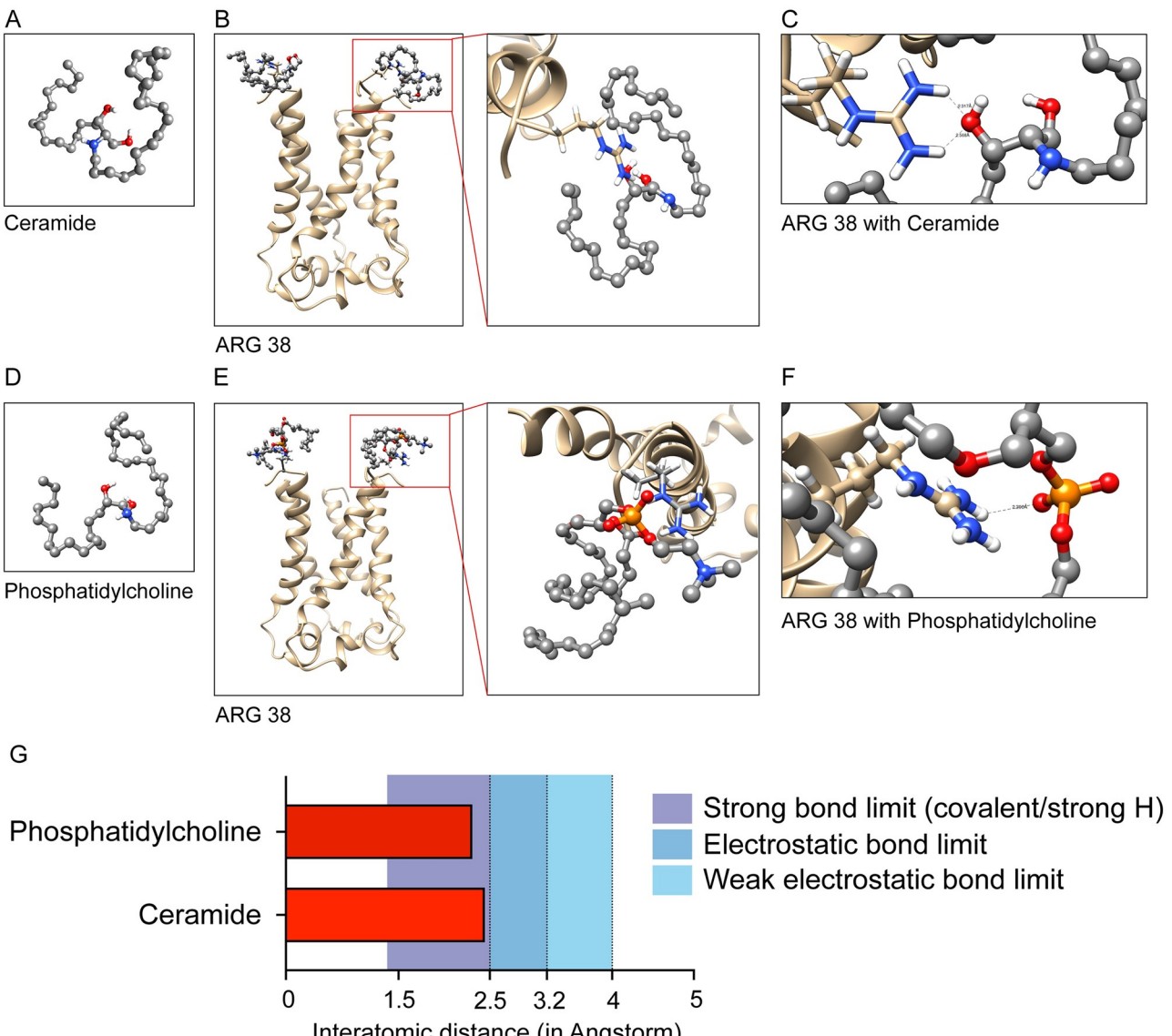

**Fig 5. E-put protein interacts with the lipid molecule components of ERGIC membrane.** (A) Ceramide: Sphingolipid generated in OpenBabel and CHIMERA. (B) Docking of Ceramide to the N-terminus of the *E-put* protein in CHIMERA-VINA and magnification of the region of interaction. (C) Interaction of ARG 38 in the C terminal with the docked ligand showing the distances between the charged groups of ARG 38 and the ligand. (D) Phosphatidylcholine: Glycerophospholipid generated in OpenBabel and CHIMERA. (E) Docking of Phosphatidylcholine to the N-terminus of the *E-put* protein in CHIMERA and Autodock Vina and magnification of the region of interaction. (F) Interaction of ARG 38 in the C terminal of *E-put* with the docked ligand showing the distances between the charged groups of ARG 38 and the ligand in CHIMERA. (G) Distances of the ligands from ARG 38 showing the bond limits of different interactions.

analyzed: (1) Closed discontinuous state (Fig 6E-i, side view, S7B-i-iii Fig) (2) Continuous channel state 1 (Fig 6E-ii, side view, S7C-i-iii) (3) Continuous channel state 2 (Fig 6E-iii, side view, S7D-i-iii Fig). We observe a particular recurring change in the PHE 26 orientation during the putative conformational changes from discontinuous closed to the continuous open state and back to the closed discontinuous state (Fig 6E-iv, side view, S7B-iii Fig). On the other hand, the orientation of the PHE 4 residue across the three conformations acts as a gate regulating the putative closed and open states of *E-put* (S7E Fig). The conformational changes during its transition from the putative closed to open states in the *E-put* protein could be a

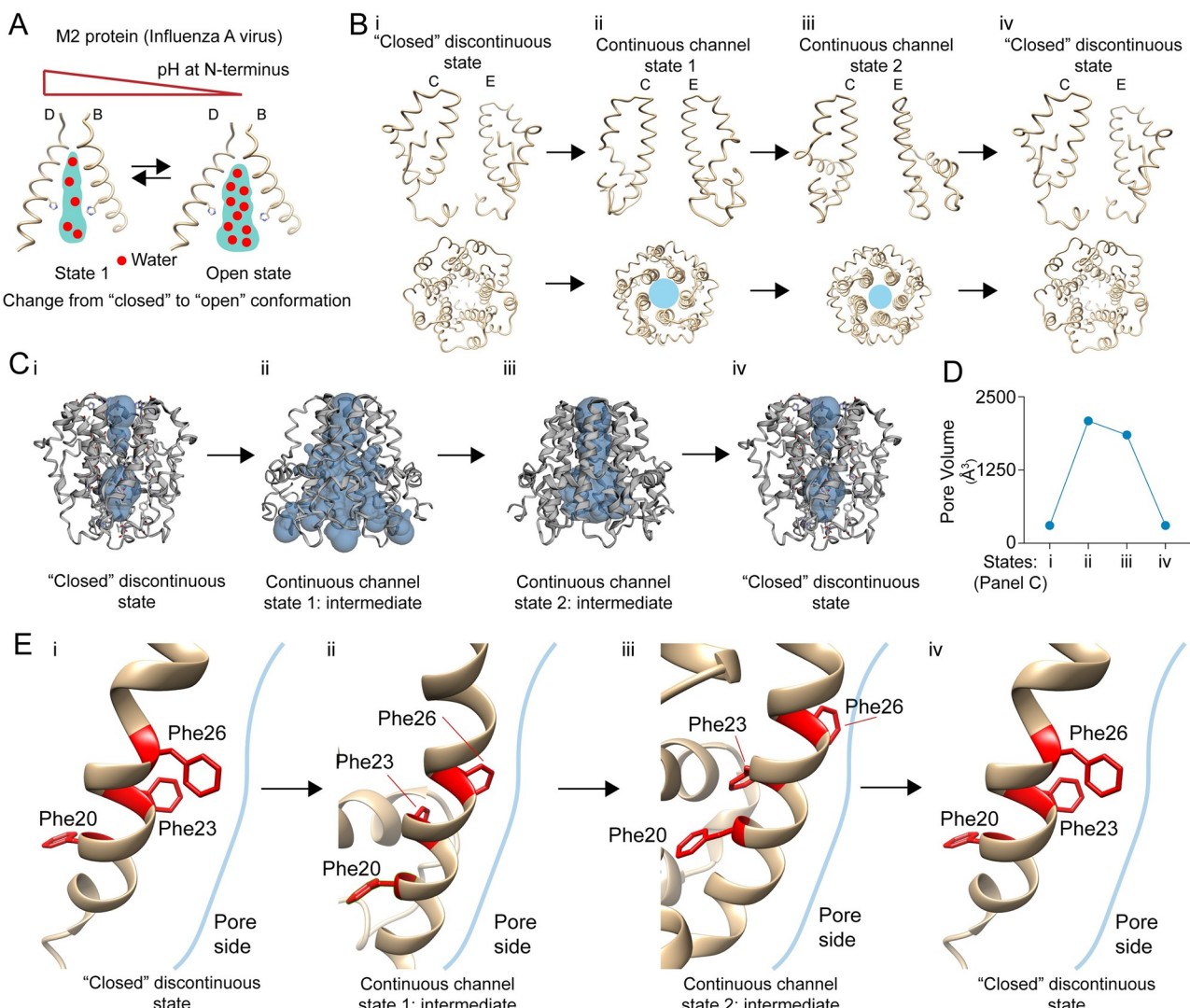

**Fig 6. Proposed mechanism of proton chaneling activity in E-put protein.** (A) Proton channeling by water hopping mechanism in M2 viroporin of influenza A virus in a pH-dependent fashion generated with CHIMERA. (B) Different conformational states of *E-put* protein in SARS-CoV-2 by CHIMERA showing its (i) Closed Discontinuous state (ii) Continuous Channel state 1 (iii) Continuous Channel state 2 (iv) Closed Discontinuous state generated with CHIMERA. (C) Pore volumes of the *E-put* protein (i) Closed Discontinuous state (ii) Continuous Channel state 1 (iii) Continuous Channel state 2 (iv) Closed Discontinuous state generated with CHIMERA and CASTp. (D) Representation of the change of pore volume as in C- i, ii, iii, iv. (E) The orientation of the bottleneck PHE 26 in different conformational states as in C- i, ii, iii, iv.

biophysical mechanism of regulation of the pore, resembling that of influenza A M2 viroporin [34].

## Insertion of *E put TM* into a lipid bilayer mimicking ERGIC membrane using CHARMM-GUI and structural morphing of the putative closed and open intermediates of *E put*

The truncated version of *E put* spans the first 40 amino acids of *E put*, a small N terminal domain (1–8), and the transmembrane pore-forming region (12–38) of the protein and would be mentioned as *E put TM* in this particular study. The glycosylated and water-solvated (defined by solvation box) closed discontinuous state (Fig 6C-i) and the continuous channel

state 1 (Fig 6C-ii) of the *E put TM* were packed with lipid-like pseudo atoms to have a general idea about the lipid packaging (Fig 7A-Closed discontinuous state, Fig 7E-Continuous channel state 1). Then the pseudo atoms were replaced by lipid components and packed the *E-put TM* individually. The composition of the lipid bilayer that mimicked the ERGIC membrane, were optimized according to Table 1. The membrane-inserted closed discontinuous state and the continuous channel state 1 of *E put TM* showed the presence of water molecules inside the pore and on two sides of the membrane (Fig 7B-Closed discontinuous state, Fig 7F-Continuous channel state 1), which corroborates with our SWISS-DOCK results. The membrane inserted *E put TM* showed that the continuous channel state 1 has a greater number of water molecules inside its pore than in the closed discontinuous state (Fig 7C-Closed discontinuous state, Fig 7G-Continuous channel state 1), which validates the existence of closed and open state conformations. We could visualize the formation of the pore from the cytoplasmic side (Fig 7D-Closed discontinuous state, Fig 7H-Continuous channel state 1). Following the insertion of the protein into the membrane and solvation, the continuous channel state 1 of *E put* was morphed using UCSF-CHIMERA taking the closed discontinuous state of *E put* as the reference. It confirmed the putative allowed conformational movements between the two structures captured at an interval of 10 frames out of the 60 interpolation steps. The possible movements of the PHE 4 at the gate region in the N terminal of the *E-put* and the PHE-26 at the bottleneck region could be visualized (Fig 7I- Frames- 1, 11, 21, 31, 41, 51, 61; S1 Video). The pore size (Fig 7J) and cross-sectional area across the entire pore (Fig 7K) of *E put TM* increased in the continuous channel state 1 (red) than in closed discontinuous state (black) demonstrating that conformation-dependent conduction of H+ ions of *E put* forming continuous water chains could be physically possible.

## Mutational basis of development of a live attenuated vaccine (LAV) and inactivated vaccine

The *E put* protein was analyzed in STRUM server by incorporating point mutations in PHE 4, GLU 8, ASN 15 and PHE 26, which might compromise its structural stability and thus result in loss of its ion channeling activity. The mutations incorporated are shown in Table 3. The mutations in the PHE 4 (gate region) and PHE 26 (bottleneck region) were introduced to increase the steric hindrance of those regions which might confer rigidity to the gate and the bottleneck respectively. This might result in loss of their movement which is necessary for the IC activity of the *E put* and lead to partial or total loss of the functionality of the protein. The mutations incorporated in the GLU 8 and ASN 15 regions should lead to the loss of the hydrogen bonding network between the water molecules and the respective residues. It will affect the formation of continuous water channels inside the pore of *E put* and might cause partial or total loss of the IC activity. Since, the IC activity of E protein is a well-studied virulent signature in SARS-CoV, so these putative loss- in- function mutations in *E put* might pave the way towards the development of live attenuated vaccine (LAV) and inactivated vaccine.

## Discussion

Structural determination of proteins using NMR spectroscopy provided a set of models of the E protein from SARS-Co-1 (Pdb id: 5X29) having the lower energies (S6A, side view, Fig 6A-ii, top view). Homology modeling of *E-put* using this template provided us with many structural models of the protein. Some of them upon proper structural refinement, represent functionally relevant structures despite having a higher Clash score or MolProbity score. The higher scores signify that the particular model is in a higher energy state correlative with functional intermediates of the protein during the viral progression.

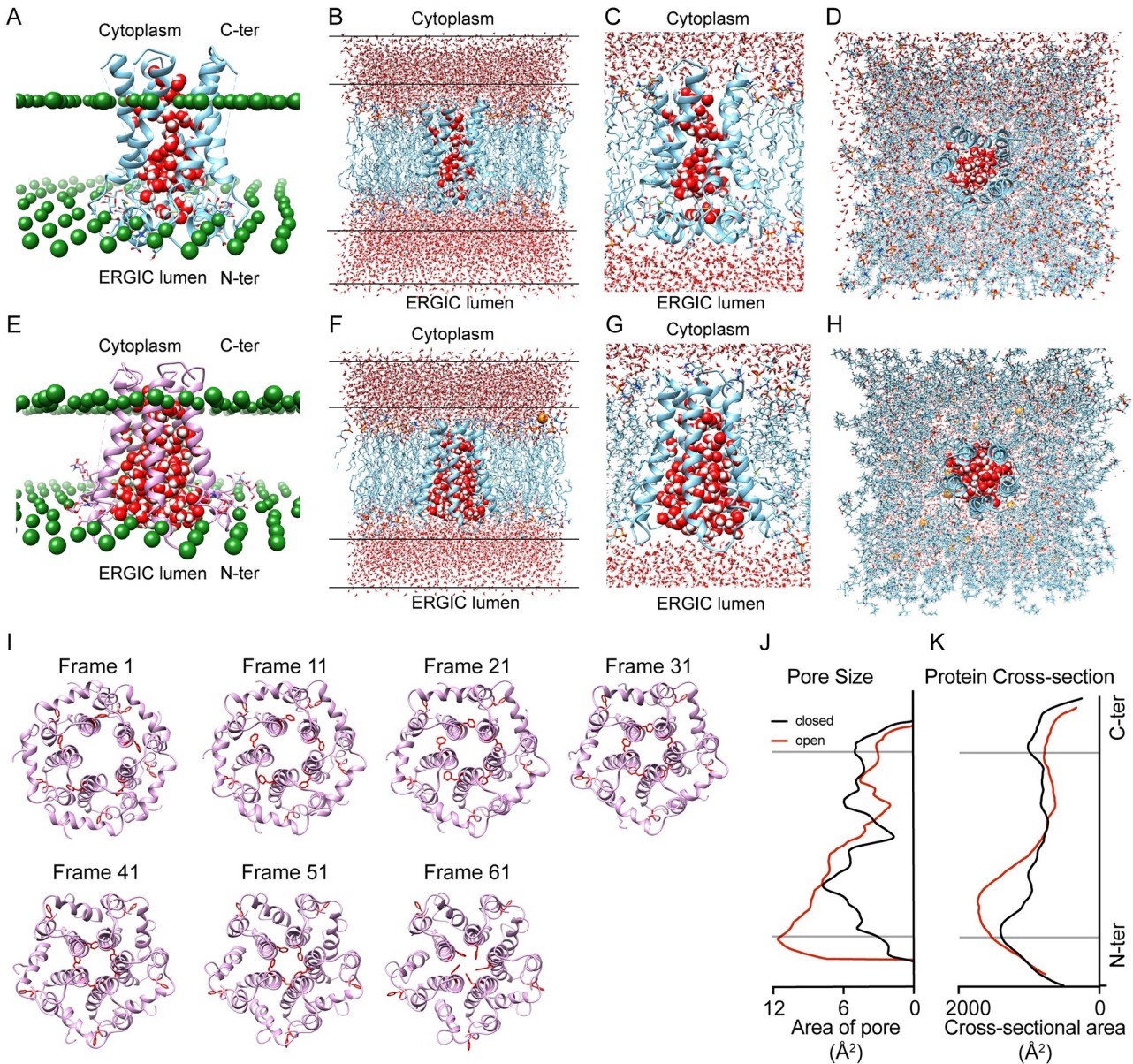

**Fig 7. Membrane insertion of *E-put* protein and structural morphing from open to closed state.** (A) Membrane insertion outline of the closed discontinuous state of *E put TM* with pore water (in red and white spheres) inside the pore structure and packed by lipid-like pseudo atoms (green spheres). Only pore water molecules remain after pore water generation through high-temperature dynamics. (B) The closed discontinuous state of *E put TM* inserted in a 1.5 lipid bilayer (POPC, POPE, POPI, POPS, Cholesterol, and SM) having 194 lipid components (94 in the upper leaflet and 100 in the lower leaflet) and modified with a TIP3P water model. The water molecules are seen in the cytoplasmic and the ERGIC luminal side. (C) A magnified side view of the model water molecules in the pore structure of the closed discontinuous state of the *E put TM* inserted in the ERGIC membrane mimic. (D) Top view of the model water molecules in the pore structure of the *E put TM* inserted in the closed discontinuous state within the ERGIC membrane mimic. (E) Membrane insertion outline of the continuous channel state 1 of *E put TM* with pore water (in red and white spheres) inside the pore structure and packed by lipid-like pseudo atoms (green spheres). Only pore water molecules remain after pore water generation through high-temperature dynamics. (F) The continuous channel state 1 of *E put TM* inserted in a 1.5 lipid bilayer (POPC, POPE, POPI, POPS, Cholesterol, and SM) having 194 lipid components (94 in the upper leaflet and 100 in the lower leaflet) and modified with a TIP3P water model. The water molecules are seen in the cytoplasmic and the ERGIC luminal side. (G) A magnified side view of the model water molecules in the pore structure of the continuous channel state 1 of the *E put TM* inserted in the ERGIC membrane mimic. (H) Top view of the model water molecules in the pore structure of the *E put TM* inserted in the continuous channel state 1 within the ERGIC membrane mimic. (I) Snapshots of the morphed protein models between open and closed state intermediate of the *E put* at a 10-frame interval (from frame 1–61). (J) Cross-sectional pore size profile of the membrane-inserted *E put TM* along the Z-axis in the closed discontinuous state (black) and continuous channel state 1 (red). (K) Cross-sectional pore area profile of the membrane-inserted *E put TM* along the Z-axis in the closed discontinuous state (close in black) and continuous channel state 1 (open in red).

**Table 3. Single site-directed mutations on the *E-put* protein.**

| Protein/Mutant | Theoretical pI (Expasy) | ddG value (STRUM) | Aliphatic index (Expasy) | Average hydrophobic score (Expasy) |
|---|---|---|---|---|
| Wild Type | 8.57 | N/A | 144.00 | 1.128 |
| F4A | 8.57 | -1.43 | 145.33 | 1.115 |
| F4L | 8.57 | -0.74 | 149.20 | 1.141 |
| F4R | 8.99 | -1.09 | 144.00 | 1.031 |
| F4Y | 8.55 | -0.6 | 144.00 | 1.073 |
| F4W | 8.57 | -0.53 | 144.00 | 1.079 |
| E8A | 8.99 | -0.53 | 145.33 | 1.199 |
| E8V | 8.99 | -0.43 | 147.87 | 1.231 |
| E8I | 8.99 | -0.66 | 149.20 | 1.235 |
| E8K | 9.26 | -0.7 | 144.00 | 1.123 |
| E8M | 8.99 | -0.55 | 144.00 | 1.200 |
| N15A | 8.57 | -0.9 | 145.33 | 1.199 |
| N15V | 8.57 | -0.67 | 147.87 | 1.231 |
| N15I | 8.57 | -0.95 | 149.20 | 1.235 |
| N15K | 8.96 | -0.68 | 144.00 | 1.123 |
| N15F | 8.57 | -0.89 | 144.00 | 1.212 |
| F26A | 8.57 | -2.14 | 145.33 | 1.115 |
| F26L | 8.57 | -0.9 | 149.20 | 1.141 |
| F26R | 8.99 | -1.18 | 144.00 | 1.031 |
| F26Y | 8.55 | -0.62 | 144.00 | 1.073 |
| F26W | 8.57 | -0.49 | 144.00 | 1.079 |

We show that the E protein of novel SARS-CoV-2 might assemble on the ER-Golgi network and ERGIC complex of human cells as pentamers similar to previous reports in other coronavirus mediated diseases like SARS-CoV-1 and MERS [33]. We have identified various component residues of the pentameric E protein lining its luminal surface which might take part in various molecular events during its conformational changes and corresponding functionalities. The ARG 38 seems to act as the hook of the protein anchoring it to the mammalian ER-Golgi membrane. The orientation of the residues of the hydrophobic funnel (LEU 37, ALA 36, ILE 33) enables water molecules to travel inside the funnel without any critical steric clashes. This is evident from the large distances of their side chains from the docked water molecule, which is beyond the H-bonding limit and short-range interactions limit as well.

The *E-put* protein has a bottleneck located at the midway of the pore and mediated by PHE 26. The bottleneck is followed by a hydrophobic pocket which we call the central core, lined by ALA 22 and LEU 18. ASN 15 and GLU 8 are the polar/charged residues present below the central pore which are at H-bond forming distance from water molecules inside the luminal surface of the pore. The N-terminal end of the *E-put* protein has a gate-like region mediated by PHE 4. The proposed intermediate structures of the *E-put* which were two of the 16 Galaxy-WEB outputs, had MolProbity and Clash score in the following order: Closed discontinuous state < Continuous Channel state 1 > Continuous Channel state 2 > Closed discontinuous state. The Continuous Channel state 1 and 2 have less stability and possibly higher energy states than the closed discontinuous state. This change of conformation of *E-put* protein is mediated by the PHE 26 at the bottleneck region which changes its orientation during the transition to the intermediates and PHE 4 which attains closed and open conformations during respective intermediates. The change in pore volume among the conformations also aligns with our hypothesis of intermediate states regulating the *E-put* protein. The *E-put TM*

(putative closed and open conformations) was inserted into an ERGIC-mimicking membrane and pore water generation along with the change of pore size and cross-sectional area of the protein validated by our hypothesis. All these *in silico* findings show that the *E-put* protein may act as an ion channel conducting $H^+$ ions. The movement of the $H^+$ ions is mediated by the formation of continuous chains of water clusters *via* a water-hopping mechanism partly similar to the M2 proton channel of influenza A virus as reported by Acharya et al., 2010 [34]. Thus, this study gives a structural insight into the putative mechanism of action of the novel SARS-CoV-2 E protein as a conformation-dependent ion channel conducting $H^+$ ions.

While there are no antiviral drugs or effective vaccines against COVID-19 on the market; we leverage the existing knowledge of viroporins and structural modeling to explain a putative mechanism involving E protein in SARS-CoV-2 related infection. The development of therapeutic approaches is being hindered due to the lack of understanding of molecular mechanisms and players responsible for this novel viral infection [5]. The E protein of the SARS-CoV has been explored as a promising vaccine candidate previously. The deletion of the E protein in a SARS-CoV (SARS-CoV-ΔE) in cell cultures or *in vivo* [25] was explored as an attenuated and effective vaccine [21, 23, 24], but it underwent a reversion and became virulent. Passage of SARS-CoV-ΔE in mice for 6 to 8 days incorporated a mutant nsp8a which is a cation selective ion channel. In cell culture, deletion of the E protein or mutation in the PDZ binding motif (PBM) at the C terminal end of the E protein, led to reversal of the virulence *via* incorporation of a novel chimeric protein with a PBM or an incorporated PBM on the E protein respectively. *In vivo* studies show that the mutant nsp-8a protein has a PBM inserted to compensate for the loss of that particular motif of the E protein. These studies show that the PBM of E protein is indispensable for viral infectivity and thus a transmembrane protein with a PBM on it can compensate for a deleted E protein. So, C-terminal truncations of the E protein, keeping the PBM intact, paved the way towards viral attenuation and development of a stable LAV. The C terminal region of the nsp 1 viral protein was deleted along with that of E protein which led to higher viral attenuation through increased host interferon responses. Viruses having both the C terminal deletions-(i) E protein and (ii) nsp 1, conferred total protection against lethal parental SARS-CoV in mice, thus making them ideal stable vaccine candidates [25]. Thus an *E-put* having function-deficient ion channel along with an intact PBM could be an ideal vaccine target. Recent studies have shed light on a large number of host-pathogen protein-protein interactions (PPIs) and identified two classes of small molecules—mRNA translation inhibitors and Sigma factor 1 and 2 receptor regulators, which showed antiviral activity [5] against novel SARS-CoV-2.

In order to gain an understanding of the E protein of novel SARS-CoV-2, we resorted to an *in-silico* approach. It is to be noted that the protein is highly hydrophobic and has a transmembrane domain which increases the difficulty of protein purification and crystallographic studies. However, they can be easily targeted for site-directed mutagenesis or deletion of a functional domain. LAV and inactivated vaccines are vaccine strategies that take advantage of the existing virulence of the virus but consists of one or more mutations or deletion of specific regions of the protein resulting in the loss of its pathogenic activity. As a result, the host is able to generate immunological memory to eliminate existing viruses and provide future protection. A possible strategy could be the deletion of the transmembrane domain (12–37) of the *E-put* protein, keeping the ER-Golgi targeting sequence in the C-terminal end and the PBM intact, which might lead to attenuation of the virus. Secondly, site-directed mutagenesis of the residues like ASN 15 and GLU 8 could also eliminate its channeling activity in accordance with the ddG values obtained from STRUM estimations. In addition, mutation of PHE 4 and PHE 26 can destabilize the structure according to the ddG values, probably by increasing rigidity to the structure. This is due to incorporation of sterically crowded residues like TRP and

TYR, which might affect its ability to change its conformation during its functionality. These mutations along with an intact PBM motif in the C terminal end of *E put* can be strategized for viral attenuation or inactivation. The knowledge of LAV development against MERS-CoV by truncations of the C terminal of the E protein along with ns1 viral protein could also be harnessed to generate similar deletion mutants of *E put* of SARS-CoV-2 *in vitro*. Viruses having the channel inactivating mutations in the transmembrane region and/or the C terminal truncations of *E put* retaining the PBM at the C terminal end could be checked for functional inactivation or attenuation, in cell culture and *in vivo* conditions. The mutational basis of the *E-put* in this study provide the platform for more rigorous and robust analyses of the protein with molecular dynamics (MD) simulations performed on the mutant protein structure. All the previously available information along with our study could open new avenues in vaccine development and other therapeutic strategies using inhibitors against the novel SARS-CoV-2.

## Supporting information

**S1 Fig.** (A) Conservation score and consensus sequence obtained from multiple sequence alignment for spike protein after a BLASTp search. (B) Conservation score and consensus sequence obtained from multiple sequence alignment for M-protein after a BLASTp search. (C) Conservation score and consensus sequence obtained from multiple sequence alignment for protein 7a after a BLASTp search. (D) Clustering showing close association of the sequences aligned in panel B and their sequence distance.
(TIF)

**S2 Fig. A-P: 16 output models from GalaxyWEB based on validation parameters from MolProbity analysis.**
(TIF)

**S3 Fig. A-P: 16 output models from SWISS-MODEL based on validation parameters from MolProbity analysis.**
(TIF)

**S4 Fig.** (A) Top view of residues' orientation of the E-put protein generated from PYMOL (i) ASN 15 (ii) GLU 8. (B) Top view of residues' orientation of the central core of the E-put protein generated from PYMOL. (C) Top view of residues' orientation of the hydrophobic funnel of the E-put protein generated from PYMOL. (D) Residues' orientation generated from PYMOL (i) side view (ii) top view showing the Gate of the E-put protein in the open conformation.
(TIF)

**S5 Fig.** (A) Superposition of the E-put and the 5X29 CoV-1 template protein (purple: E-put; green: 5X29) as (i) side view and (ii) top view. (B) Superposition of the SWISS-MODEL and the 5X29 CoV-1 template protein (green: 5X29; cyan: SWISS-MODEL) as (i) side view and (ii) top view.
(TIF)

**S6 Fig.** (A) Structure ensembles of the 5X29 NMR structures as (i) side view and (ii) top view. (B) Ramachandran plot of the 5X29 NMR structure of the CoV-1protein (template).
(TIF)

**S7 Fig.** (A) Docking of water to the modeled structure of E-put protein in its different conformations (i) Closed discontinuous state (ii) Continuous channel state 1 (iii) Continuous channel state 2. (B) (i-ii) Top view showing all five bottleneck phenylalanine residues (PHE 26)

along with adjacent phenylalanine residues (PHE 23 and PHE 20) for the closed discontinuous state. (iii) Enlarged image of the top view orientation of the PHE 26, PHE 23, and PHE 20 on the E chain. (C) (i-ii) Top view showing all five bottleneck phenylalanine residues (PHE 26) along with adjacent phenylalanine residues (PHE 23 and PHE 20) for the open continuous channel state 1. (iii) Enlarged image of the top view orientation of the PHE 26, PHE 23, and PHE 20 on the E chain. (D) (i-ii) Top view showing all five bottleneck phenylalanine residues (PHE 26) along with adjacent phenylalanine residues (PHE 23 and PHE 20) for the continuous channel state 2. (iii) Enlarged image of the top view orientation of the PHE 26, PHE 23, and PHE 20 on the E chain. (E) The orientation of the PHE 4 residues of the Gate of the E protein generated in CHIMERA in its closed and open conformational state.
(TIF)

**S1 Video. Structural morphing from open to closed state of the *E-put* protein.** A movie showing the possible intermediates morphed between the open continuous channel state and the closed discontinuous channel state of the *E put* protein.
(MP4)

## Acknowledgments

We thank Mr. Jaydeep Paul for insightful discussions on the research idea, Prof. Félix Rey, Dr. Aleksandra Polosukhina and Dr. Georgios Agoranos for reading the manuscript, suggesting additions and changes, Ms. Madhuparna Chakraborty for proof-reading the manuscript. We earnestly thank Prof. Pierre-Marie Lledo and Prof. Pinak Chakrabarti for providing encouragement to conduct the work and reading the finished manuscript.

## Author Contributions

**Conceptualization:** Manish Sarkar, Soham Saha.

**Formal analysis:** Manish Sarkar, Soham Saha.

**Investigation:** Manish Sarkar, Soham Saha.

**Methodology:** Manish Sarkar, Soham Saha.

**Project administration:** Soham Saha.

**Software:** Manish Sarkar, Soham Saha.

**Writing – original draft:** Manish Sarkar, Soham Saha.

**Writing – review & editing:** Manish Sarkar, Soham Saha.

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
