## [Decision Letter · Decision Letter 0]

18 Jun 2020

PONE-D-20-16258

Structural insight into the role of novel SARS-CoV-2 E protein: Potential target for LAV development and other therapeutic strategies

PLOS ONE

Dear Dr. Saha,

Thank you for submitting your manuscript to PLOS ONE. After careful consideration, we feel that it has merit but does not fully meet PLOS ONE’s publication criteria as it currently stands. Therefore, we invite you to submit a revised version of the manuscript that addresses the points raised during the review process.

We look forward to receiving your revised manuscript.

Kind regards,

Yang Zhang

Academic Editor

PLOS ONE

Journal Requirements:

3. We note you have included a table to which you do not refer in the text of your manuscript. Please ensure that you refer to Table 1 in your text; if accepted, production will need this reference to link the reader to the Table.

Reviewers' comments:

Reviewer's Responses to Questions

**Comments to the Author**

1. Is the manuscript technically sound, and do the data support the conclusions?

Reviewer #1: Yes

Reviewer #2: Partly

Reviewer #3: Yes

2. Has the statistical analysis been performed appropriately and rigorously? 

Reviewer #1: Yes

Reviewer #2: N/A

Reviewer #3: Yes

3. Have the authors made all data underlying the findings in their manuscript fully available?

Reviewer #1: Yes

Reviewer #2: Yes

Reviewer #3: Yes

4. Is the manuscript presented in an intelligible fashion and written in standard English?

Reviewer #1: Yes

Reviewer #2: Yes

Reviewer #3: Yes

5. Review Comments to the Author

Reviewer #1: In the manuscript, the authors found that E proteins demonstrate lower disparity and mutability. In addition, the authors determined that GLU 8 and ASN 15 in the N terminal region were in close proximity to form H-bonds. The authors also proposed a mechanism of viral proton channeling activity which may play a critical role in viral infection. The manuscript is interesting and well-written. This reviewer feels that the manuscript can be accepted in the current form.

Reviewer #2: This manuscript described an in silico study on identifying the structural insights into the role of the SARS-CoV-2 envelope (E) protein, which may be a target for LAC development. So far most studies are focused on the spike protein and/or the main protease for RNA replication. The study provides a different perspective on the vaccine development by focusing on the E protein. This work itself is interesting, but in my opinion the manuscript still has some weaknesses.

Major comments:

1. The manuscript focused too much on the techniques and details, and this to some extent weaken the scientific logic. Although the possibility of E-protein being a vaccine candidate has been explored with the MERS-CoV, I think it is quite necessary to clarify and highlight this point. The authors dedicated a lot of space on describing and calculating the ion channel, but they did not show how the ion channel will affect the infectivity and virulence of SARS-CoV-2.

2. How the docking with water molecules is performed, because there are many water molecules shown in Fig. 3 and 4. It looks like a molecular dynamics simulation to me, but clearly it was regarded as molecular docking with SWISS-DOCK in the manuscript. The authors mentioned that they have successfully modelled the opening and closing conformations based on homology modeling. In my opinion, this is totally determined by the template they used (i.e. 5X29); in 5X29, there are 16 solution models, and therefore they modelled 16 structures using GalaxyWEB and SWISS-MODEL, respectively. Suppose there is no NMR structure but an X-ray structure that can be used as a template, I wonder whether they can still model the opening/closing conformations. In this context, I think an MD simulation is probably necessary.

3. The authored mentioned that “LAV is a vaccine strategy that takes advantage of the existing virulence of the virus but consists of one or more mutations or deletion of specific regions of the protein resulting in the loss of its pathogenic activity”. I wonder whether there is a way to figure out ALL the appropriate mutation positions. The authors suggested that “Secondly, site-directed mutagenesis of the residues like ASN 15 and GLU 8 could also eliminate its channeling activity. In addition, mutation of PHE 4 and PHE 26 can render rigidity to the structure affecting its ability to change its conformation during its functionality.” I would like the authors to provide some computational evidence that the mutations may eliminate the channeling activity. Can the mutation assessment tools like STRUM (Lijun Quan, Qiang Lv, Yang Zhang. STRUM: Structure-based stability change prediction upon single-point mutation, Bioinformatics, 32: 2911-19 (2016)), SSIPe (https://doi.org/10.1093/bioinformatics/btz926) and EvoEF2 (https://doi.org/10.1093/bioinformatics/btz740) be used for quantifying the role of these mutations? For example, how the mutations affect the binding or protein stability? I think it would be more interesting and attractive to readers by performing some in silico mutagenesis studies rather than just focusing on the structure modeling and characterization, because the mutations may be useful suggestions for LAV development.

Minor comments:

1. “the E protein from SARS-CoV-1 (Pdb id: 5X29) having the minimum energies”. I do not think all of the set of NMR models are having minimum energies, but I agree that they have low energies.

2. “All these in silico findings show that the E-put protein may act as a proton channel.” I do not think a “proton channel” is equal to an “ion channel”.

3. It is better to change the background color in the consensus logo figures from black to white (e.g. Fig. 1 and Supplementary Fig. S1), because it cannot be clearly distinguished by eyes.

Reviewer #3: The manuscript is interesting. The authors may provide more discussion on the comparison of the MERS LAV that incorporated the truncation near C-terminal. Furthermore, the results of the in-silico work should be discussed in context of any available experimentally-derived information about the putative proton channel and the effect of C-terminal mutations on the pathogenicity of the SARS-CoV-2 virus.

6. PLOS authors have the option to publish the peer review history of their article (what does this mean?). If published, this will include your full peer review and any attached files.

Reviewer #1: No

Reviewer #2: No

Reviewer #3: No

---

## [Author Response · Author response to Decision Letter 0]

15 Jul 2020

Detailed responses:

Below we have listed verbatim the concerns followed by our responses in blue.

Please note that the line and page numbers in the following rebuttal correspond to the manuscript version “without track changes”.

Journal Requirements:

We have revised the format of the manuscript and followed the PLOS One style requirements as suggested.

We would like to change the status of the data availability. All relevant data are in the manuscript text in the form of tables, figures and supplementary information. The ‘.pdb’ models of the proteins generated in this study are available in the Github account for download:

https://github.com/SohamSahaNeuroscience/Covid-19_Eprotein_models

We have provided all structural modeling data in the manuscript and in the Github link.

https://github.com/SohamSahaNeuroscience/Covid-19_Eprotein_models

b) If there are no restrictions, please upload the minimal anonymized data set necessary to replicate your study findings as either Supporting Information files or to a stable, public repository and provide us with the relevant URLs, DOIs, or accession numbers. For a list of acceptable repositories, please see

http://journals.plos.org/plosone/s/data-availability#loc-recommended-repositories.

The Github link to the ‘.pdb’ files of the models are available at:

https://github.com/SohamSahaNeuroscience/Covid-19_Eprotein_models

3. We note you have included a table to which you do not refer in the text of your manuscript. Please ensure that you refer to Table 1 in your text; if accepted, production will need this reference to link the reader to the Table.

We have included 2 more tables. Table 1 is cited in Page 12, Line 257. Table 2 (formerly Table 1) is cited in Page 17, Lines 363-364. Table 3 is cited in Page 27 Line 595.

Review Comments to the Author

Reviewer #1: In the manuscript, the authors found that E proteins demonstrate lower disparity and mutability. In addition, the authors determined that GLU 8 and ASN 15 in the N terminal region were in close proximity to form H-bonds. The authors also proposed a mechanism of viral proton channeling activity which may play a critical role in viral infection. The manuscript is interesting and well-written. This reviewer feels that the manuscript can be accepted in the current form.

We thank the Reviewer for the encouraging comment. However, as suggested by other reviewers, we have expanded on the introduction, results and discussion of activity of viral ion channels conducting H+ ions and their role in viral infection.

Reviewer #2: This manuscript described an in silico study on identifying the structural insights into the role of the SARS-CoV-2 envelope (E) protein, which may be a target for LAV development. So far most studies are focused on the spike protein and/or the main protease for RNA replication. The study provides a different perspective on the vaccine development by focusing on the E protein. This work itself is interesting, but in my opinion the manuscript still has some weaknesses.

Major comments:

1. The manuscript focused too much on the techniques and details, and this to some extent weaken the scientific logic. Although the possibility of E-protein being a vaccine candidate has been explored with the MERS-CoV, I think it is quite necessary to clarify and highlight this point. The authors dedicated a lot of space on describing and calculating the ion channel, but they did not show how the ion channel will affect the infectivity and virulence of SARS-CoV-2.

We agree with the reviewer on this comment. In order to drive the scientific logic, we have expanded our introduction to review where the study falls in the light of the information available. We have introduced in greater detail, what role a viroporin plays in viral pathogenesis, how it can impact the normal functioning of the host cells and how they can be potential targets for therapeutics. In view of the existing literature, we show that the E-protein in SARS-CoV2 is a robust viroporin forming pentameric structures with an H+ ion conductive pore, and extensive study of this protein might lead to strategies to counteract the existing threat. The scientific concepts we added in the introduction are summarized as follows:

A. Role of viroporins: Page 3, Line 62-66; Page 5, Lines 83-84. 

B. E protein as a vaccine candidate in MERS-CoV: Page 3-4, Lines: 66-82.

C. Mutational studies and observations from SARS-CoV-1: Page 4-5, Lines 85-96.

D. E protein ion channeling activity: Page 5, Lines 98-106.

E. New in silico observations in our studies: Page 6, Lines 119-120.

A new paragraph has been added in the section “Discussion” to discuss our findings with respect to the existing reports in the literature. Insertion of the E protein of the SARS-CoV-2 in an ERGIC membrane along with solvation and pore water generation using CHARMM-GUI corroborates with our previous findings. We discussed the challenges like virulence reversal in the vaccine development pipeline which have been counteracted previously. The E-protein of SARS-CoV-2 having a mutant TM with an intact PBM have been proposed to be an ideal vaccine target on Page 30-31, Lines 655-679. A further discussion regarding this potential of the E protein is SARS-CoV-2 have been included on Page 32, Lines 699-708.

2. How the docking with water molecules is performed, because there are many water molecules shown in Fig. 3 and 4. It looks like a molecular dynamics simulation to me, but clearly it was regarded as molecular docking with SWISS-DOCK in the manuscript. The authors mentioned that they have successfully modelled the opening and closing conformations based on homology modeling. In my opinion, this is totally determined by the template they used (i.e. 5X29); in 5X29, there are 16 solution models, and therefore they modelled 16 structures using GalaxyWEB and SWISS-MODEL, respectively. Suppose there is no NMR structure but an X-ray structure that can be used as a template, I wonder whether they can still model the opening/closing conformations. In this context, I think an MD simulation is probably necessary.

Water docking on the E-protein was performed by SWISS-DOCK blind docking algorithm. As the Reviewer suggested, we have now clarified the observation by membrane insertion of the pentameric E-protein into a closely mimicked ERGIC membrane using CHARMM-GUI. In addition, we show that both the open and closed conformation intermediates can insert into the membrane with water molecules inside the pore similar to SWISS DOCK results with the exception of the membrane occupied region which got clarified from CHARMM-GUI results. We also used a morphing protocol in UCSF-CHIMERA to predict the possible intermediate states between the open and closed conformation.

We agree with the reviewer that a MD simulation is probably necessary. In order to address the question, we resorted to NAMD and VMD using the outputs from CHARMM-GUI and other additional parameter files from CHARMM-GUI Archive, to run a simulation of 106 steps each of 1 fs having an acquisition frequency of 500 steps. We performed the simulation study at 303.18 K and 1.018 bar atmospheric pressure. There are several other parameters in NAMD which can be regulated to optimize such a study but lack of higher computational power hindered our study just before the optimization step. Lockdown in India, especially in Kolkata, did not allow us (Manish Sarkar) to get access to higher computational systems of my institute, which has been totally shut down with no further notice. In France, we (Soham Saha) couldn't get institutional access to the supercomputer facilities due to disruption of normal functioning of laboratory activities, which has only returned to normalcy. 

The added changes in the manuscript (section Methods) are seen here:

A. Methods: Membrane insertion using CHARMM-GUI: Page 11-12, Lines 242-270.

B. Table1: composition and optimization of the generated membrane: Page 12, Line 271.

C. Methods: Morphing using open and closed conformations: Page 13, Lines 274-286.

The added changes in the manuscript (section Results) are seen here:

A. Insertion of E-put TM in ERGIC-membrane: Pages 24-25, Lines 530-558.

B. We have added a new figure: Fig 7: Page 25-26, Lines 559-589 (figure legend).

C. We have added a new video of the morph between closed and open conformations of the protein to account for the possible intermediates during the change. Video 1: Page 27, Line 587-589.

3. The authored mentioned that “LAV is a vaccine strategy that takes advantage of the existing virulence of the virus but consists of one or more mutations or deletion of specific regions of the protein resulting in the loss of its pathogenic activity”. I wonder whether there is a way to figure out ALL the appropriate mutation positions. The authors suggested that “Secondly, site-directed mutagenesis of the residues like ASN 15 and GLU 8 could also eliminate its channeling activity. In addition, mutation of PHE 4 and PHE 26 can render rigidity to the structure affecting its ability to change its conformation during its functionality.” I would like the authors to provide some computational evidence that the mutations may eliminate the channeling activity. Can the mutation assessment tools like STRUM (Lijun Quan, Qiang Lv, Yang Zhang. STRUM: Structure-based stability change prediction upon single-point mutation, Bioinformatics, 32: 2911-19 (2016)), SSIPe (https://doi.org/10.1093/bioinformatics/btz926) and EvoEF2 (https://doi.org/10.1093/bioinformatics/btz740) be used for quantifying the role of these mutations? For example, how the mutations affect the binding or protein stability? I think it would be more interesting and attractive to readers by performing some in silico mutagenesis studies rather than just focusing on the structure modeling and characterization, because the mutations may be useful suggestions for LAV development.

We used STRUM and ExPasy servers to explore the effects of single point mutations on the important residues that we discussed in the manuscript. We thank the reviewer for suggesting STRUM to validate our mutations depending on their ddG values which could serve as a platform for identifying loss-in-function or non-functional mutants which would shed light to the potential in vitro and in vivo studies. In this process, mutations can be tested in cell cultures and/or animal models and could be used for the development of therapeutic and vaccine strategies.

The added changes in the manuscript (section Methods) are seen here:

A. Methods: Point mutations using ProtParam and ExPasy: Page 13-14, Lines 288-302.

The added changes in the manuscript (section Results) are seen here:

A. Mutational basis of vaccine development: Page 27, Lines 591-605.

B. Table showing effects of site directed mutagenesis: Table 3, Page 28, Line 610.

Minor comments:

1. “the E protein from SARS-CoV-1 (Pdb id: 5X29) having the minimum energies”. I do not think all of the set of NMR models are having minimum energies, but I agree that they have low energies.

We have changed the phrase from “minimum energies” to “lower energies” in Page 29; Line 616.

2. “All these in silico findings show that the E-put protein may act as a proton channel.” I do not think a “proton channel” is equal to an “ion channel”.

The ion conductivity of E protein has also been elucidated in CoVs from other genera. Synthetic E protein showed slight preference in conduction of cations over anions in reconstituted lipid membranes with composition and charge mimicking the ERGIC membrane, and thus showed no specific selectiveness towards any cation. We incorporated the literature in the introduction (Page 4, Lines 83-88).

We modified the term “proton channel” to “ion channels conducting H+ ions”:

a. Page 2, Line 29

b. Page 18, Line 381

c. Page 18, Line 388

d. Page 25, Line 558

e. Page 30, Line 650

f. Page 30, Line 654

3. It is better to change the background color in the consensus logo figures from black to white (e.g. Fig. 1 and Supplementary Fig. S1), because it cannot be clearly distinguished by eyes.

We changed the background color from black to white, as suggested by the reviewer in Fig 1 and Supplementary Fig 1.

Reviewer #3: The manuscript is interesting. The authors may provide more discussion on the comparison of the MERS LAV that incorporated the truncation near C-terminal. Furthermore, the results of the in-silico work should be discussed in context of any available experimentally-derived information about the putative proton channel and the effect of C-terminal mutations on the pathogenicity of the SARS-CoV-2 virus.

We thank the reviewer for the encouraging comment. We acknowledge that our discussion of the vaccine with respect to the MERS LAV with truncation near the C-terminal region has not been discussed. The MERS LAV approach is now incorporated in the introduction: Page 3-4, Lines: 66-82.

A new paragraph has been added in the section “Discussion” to discuss our findings with respect to the existing reports in the literature. We discussed the challenges and new therapeutic potential of the E-protein on Page 30-31, Lines 655-679. A further discussion has been included on Page 32, Lines 699-708.

A. “truncation near C-terminal effect of C-terminal mutations on pathogenicity”: We have discussed this point in Page 30-31 Lines 659-679.

B. “putative proton channel”: We modified the term “proton channel” to “ion channels conducting H+ ions”. We discuss this point in Page 4 Lines 83-87

Further, we discuss the effect of potential point mutations in the E protein of SARS-CoV-2.

We used STRUM and ExPasy servers to explore the effects of single point mutations on the important residues that we discussed. 

The added changes in the manuscript (section Methods) are seen here:

Methods: Point mutations using ProtParam and ExPasy: Page 13-14, Lines 288-302.

The added changes in the manuscript (section Results) are seen here:

A. Mutational basis of vaccine development: Page 27, Lines 591-605.

B. Table showing effects of site directed mutagenesis: Table 3, Page 28, Line 610.

We propose SARS-CoV-2 with engineered E protein as an ideal vaccine strategy on Page 33 Lines 708-714.

---

## [Decision Letter · Decision Letter 1]

27 Jul 2020

Structural insight into the role of novel SARS-CoV-2 E protein: A potential target for vaccine development and other therapeutic strategies

PONE-D-20-16258R1

Dear Dr. Saha,

We’re pleased to inform you that your manuscript has been judged scientifically suitable for publication and will be formally accepted for publication once it meets all outstanding technical requirements.

Kind regards,

Yang Zhang

Academic Editor

PLOS ONE

Additional Editor Comments (optional):

Reviewers' comments:

Reviewer's Responses to Questions

**Comments to the Author**

1. If the authors have adequately addressed your comments raised in a previous round of review and you feel that this manuscript is now acceptable for publication, you may indicate that here to bypass the “Comments to the Author” section, enter your conflict of interest statement in the “Confidential to Editor” section, and submit your "Accept" recommendation.

Reviewer #2: All comments have been addressed

2. Is the manuscript technically sound, and do the data support the conclusions?

Reviewer #2: Yes

3. Has the statistical analysis been performed appropriately and rigorously? 

Reviewer #2: Yes

4. Have the authors made all data underlying the findings in their manuscript fully available?

Reviewer #2: Yes

5. Is the manuscript presented in an intelligible fashion and written in standard English?

Reviewer #2: Yes

6. Review Comments to the Author

Reviewer #2: (No Response)

7. PLOS authors have the option to publish the peer review history of their article (what does this mean?). If published, this will include your full peer review and any attached files.

Reviewer #2: No

---

## [Editor Report · Acceptance letter]

30 Jul 2020

PONE-D-20-16258R1 

Structural insight into the role of novel SARS-CoV-2 E protein: A potential target for vaccine development and other therapeutic strategies 

Dear Dr. Saha:

I'm pleased to inform you that your manuscript has been deemed suitable for publication in PLOS ONE. Congratulations! Your manuscript is now with our production department. 

Kind regards, 

on behalf of

Dr. Yang Zhang 

Academic Editor

PLOS ONE